# Influence of Secondary Metabolites According to Maturation of Perilla (*Perilla frutescens*) on Respiratory Protective Effect in Fine Particulate Matter (PM2.5)-Induced Human Nasal Cell

**DOI:** 10.3390/ijms252212119

**Published:** 2024-11-12

**Authors:** Min-Young Kim, Jung-In Kim, Sang-Woo Kim, Sungup Kim, Eunyoung Oh, Jeongeun Lee, Eunsoo Lee, Yeon-Ju An, Chae-Yeon Han, Heungsu Lee, Myoung-Hee Lee

**Affiliations:** Department of Southern Area Crop Science, National Institute of Crop Science, Rural Development Administration, Milyang 50424, Republic of Korea; kji1204@korea.kr (J.-I.K.); kimsw1021@korea.kr (S.-W.K.); sesameup@korea.kr (S.K.); lavondy10@korea.kr (E.O.); jel6418@korea.kr (J.L.); awesomelee@korea.kr (E.L.); ayj3043@korea.kr (Y.-J.A.); hcy4246@korea.kr (C.-Y.H.); lhs2khk@korea.kr (H.L.); emhee@korea.kr (M.-H.L.)

**Keywords:** perilla flower, phenolics, growth periods, respiratory disease, fine particulate matter

## Abstract

Fine particulate matter (PM2.5) exposure worsens chronic respiratory diseases through oxidative stress and inflammation. *Perilla frutescens* (L.) has potential respiratory protective properties, but the impact of growth stages on its beneficial metabolites is unclear. We aimed to evaluate how different growth stages affect phenolic acids, flavonoids, and polycosanols in perilla seeds and flowers and their efficacy in countering PM2.5-induced damage. Perilla seeds and flowers from five varieties at 10, 20, 30, and 40 days post-flowering were analyzed for metabolite content. Their antioxidant, anti-inflammatory, and respiratory protective effects were tested in RPMI 2650 cells. Our findings indicated that perilla flowers contained higher levels of functional components than seeds and exhibited significant variation with maturation. Phenolic acids of perilla flowers were highest at the early stages of maturation after flowering. However, individual flavones of perilla flowers were the highest at the late maturation stages after flowering. Extracts from perilla flowers harvested 20 days after flowering exhibited significant respiratory protection, effectively inhibiting inflammatory cytokines, mucus secretion, and oxidative stress markers. In conclusion, the flower parts of perilla, particularly those harvested 20 days after flowering, are useful materials for obtaining phenolic compounds, including rosmarinic acid, with high antioxidant and respiratory enhancement effects.

## 1. Introduction

Chronic respiratory diseases have increased prevalence and mortality due to air pollution, which is strongly associated with exposure to fine particulate matter with an aerodynamic diameter of less than 2.5 μm (PM2.5) [1]. Fine particulate matter exposure worsens symptoms in patients with chronic obstructive pulmonary disease (COPD) [2,3], and extensive research has documented the link between air pollution and COPD [4]. Long-term exposure to PM2.5 causes emphysematous lesions, increases airway inflammation, and reduces lung function in COPD patients [5].

Generally, PM2.5 exposure triggers oxidative stress and inflammation throughout the nasal, bronchial, and pulmonary airways [6]. Increased mRNA expression and protein secretion of interleukin (IL)-1α, IL-1β, IL-6, IL-8, and tumor necrosis factor-alpha (TNF-α) in epithelial cells following PM2.5 exposures have been seen in numerous in vitro and in vivo investigations [7]. Exposure to air pollutants causes increased expression of TSLP and IL-33, accelerating T2-dependent airway inflammation [8,9]. In addition to increasing the activity of matrix metalloproteinases (MMPs) in epithelial cells, fine particulate matter stimulation also causes downregulation of E-cadherin, b-catenin, or claudin, which is a factor in the remodeling and malfunction of epithelial cells [10]. Prolonged exposure to air pollution also promotes mucus hypersecretion and airway remodeling [11]. Thus, epithelial dysfunction, resulting from these pollution-induced processes, plays a key role in the obstructive pathophysiology associated with air pollution. Therefore, developing natural products and new drugs is crucial to protect airway cells from air-pollution-induced fine particulate matter and reduce airway inflammation.

*Perilla frutescens* (L.), a traditional herbal medicine, has been widely used in East Asia for treating colds, coughs, nausea, vomiting, abdominal pain, constipation, asthma, and food poisoning [12]. It is considered a herbal remedy due to its potent antioxidant, anticancer, antibacterial, and antiallergic properties, which are largely attributed to its secondary metabolites. Hou et al. [12] reported several secondary metabolites in perilla seeds, leaves, stems, and flowers, such as phenolic acids, flavonoids, glycosides, alkaloids, triterpenes, and policosanols, which may contribute to its health benefits.

Rosmarinic acid (RA), a major phenolic acid in perilla, has demonstrated antioxidant and anti-inflammatory effects, including respiratory protection. It targets β2-adrenergic receptors and offers antiasthmatic effects by inhibiting the NF-κB signaling pathway [13]. The main flavonoids like luteolin, apigenin, and their glycosides have shown potential benefits for respiratory disorders by reducing bronchial inflammation [14] and inhibiting the influenza virus [15]. Although studies have explored the respiratory protective effects of perilla extracts and their secondary metabolites, the impact of different growth stages on the biosynthesis of these metabolites and their respiratory health benefits remains unclear.

The biosynthesis and accumulation of secondary metabolites in plants are included in factors such as genetic traits, growth period, flowering time, and growth conditions [8,9,16]. These metabolites are synthesized in all plant parts, with their concentrations varying throughout plant growth and development. This variation is mainly due to changes in the expression of genes that regulate enzyme activity involved in the biosynthesis of phenolic compounds, which can be affected by both genetic traits and environmental factors [17]. Phenolic acids, such as RA and caffeic acid, play a crucial role in the biosynthesis of various secondary metabolites through the shikimate pathway [18]. Therefore, understanding the growth periods when this pathway is most active is important for optimizing functional metabolite production.

Secondary metabolites from all parts of the perilla plant are considered potential materials for respiratory protection from pollutants. In various other medicinal plants, such as rosemary, biosynthesis of phenolic compounds has been reported to vary with maturation after flowering time [19]. These variations can be influenced by growth stages, geographical origin, maturity, genetic factors, utilization part of the plant, postharvest drying, and storage conditions [20]. However, information on the differences in the biosynthesis of secondary metabolites according to the utilized parts, growth stages, and maturity of perilla is insufficient, and studies on the influence of these variations in secondary metabolites on health benefits, including respiratory protective effects, are lacking. Therefore, the objectives of this study are to (a) investigate phenolic acid, flavonoid, and polycosanol in perilla seeds and flowers with five varieties according to growth periods (10, 20, and 30 d) after flowering, and (b) evaluate their PM2.5-induced respiratory protection, anti-inflammatory, and inhibitory effects on mucus hypersecretion and fibrosis in RPMI 2650 cells.

## 2. Results and Discussion

### 2.1. Flowering Stage, Extraction Yield, Total Polyphenol, and Flavonoid Content

Perilla (*Perilla frutescens*) is a typical short-day plant, with flowering occurring when daylight hours are reduced from approximately 12 h or less [21]. Flowers are crucial plant organs, essential for plant reproduction and evolutionary growth [22]. The period from bud formation to flowering is the most active growth event during flower development [22], accompanied by important changes in primary and secondary metabolites and various physiological processes in plants. These changes in metabolites during flower development can affect the nutritional and health-promoting properties of plants [23].

As shown in Figure 1, the flowering stages of Dayu, Anyu, Milyang 78, Milyang 90, and YCPL243 were observed on 5 September, 25 August, 2 September, 30 August, and 6 September, respectively. The Anyu cultivar (25 August) had the earliest flowering stage and the YCPL243 genetic resource (6 September) had the latest, with a difference of up to 11 d. Different cultivars and genetic resources show varying sensitivities to daylight, resulting in a wide variation in flowering stages [24]. Based on the research by Kim et al. [25], which showed that the grain-filling period, or the period from flowering to maturity, tends to be 30–34 days depending on the variety, we decided to harvest 40 days after the flowering stage. Dayu, Anyu, Milyang 78, Milyang 90, and YCPL243 were harvested between 10 to 40 days after flowering stages. Samples were collected from 5 September to 15 October, 4 September to 4 October, 2 September to 12 October, 30 August to 9 October, and 6 September to 16 October, respectively. The samples were freeze-dried, and extracts were prepared to investigate the accumulation of secondary metabolites and their physiological characteristics.

Flavonoid-rich extracts (FRE) from perilla seeds and flowers of different varieties and growth periods were prepared using an optimized method with 60% ethanol to extract residues from freeze-dried samples harvested at various stages after flowering (Figure 2, Appendix A). The extraction yield was generally higher in perilla flowers (4.20–13.18%) than those in perilla seeds (3.29–7.27%), regardless of variety and growth period. Moreover, perilla flowers showed significant variation in yield across growth periods, unlike the seeds. Yield in flowers ranged from 8.58 to 9.73% at 10 days after flowering, increased to 9.90–13.18% at 20 days, but then decreased, reducing to 4.41–5.88% by 40 days. Similar trends of decreased extraction yield during the late blooming stage have been observed in previous studies on the aerial parts of various plants, including amaranth, flax, sunflower, and soy [26].

These phenolic compounds in these extract residues were comprehensively analyzed, and the total polyphenol contents (TPC) and flavonoid contents (TFC) of perilla seeds and flowers across different varieties and growth periods are shown in Figure 2, Appendix A. TPCs in perilla seed ranged from 5.62 to 8.46 mg of gallic acid equivalent (GAE)/g extract residue (ER) and did not differ significantly among the extracts obtained from seeds at different growth stages compared to perilla flower. In contrast, TPCs in perilla flowers were higher, ranging from 6.13 to 22.56 mg GAE/g ER, and showed considerable variation depending on maturation.

Similar to the trends of extraction yields and TPC results, TFCs were generally higher in perilla flowers (4.47–33.71 mg CE/g ER) than in perilla seeds (5.10–9.67 mg CE/g ER), regardless of variety and growth period. TPCs and TFCs varied significantly with maturation. At 10 days post-flowering, TPC ranged from 17.88 to 26.02 mg GAE/g ER, and TFC from 16.63 to 89.89 mg CE/g ER. These values increased at 20 days post-flowering but showed a decreasing trend after 30 days, reducing to 6.13–8.00 mg GAE/g ER and 4.47–7.66 mg CE/g ER by 40 days. This suggests that phenolic compounds are more abundant in flower buds and seed coats than in seeds. As the proportion of seeds increases with plant maturation after flowering, the extracts contain a higher proportion of sugars, carbohydrates, and soluble proteins. This phenomenon has been observed in other crops and fruits, such as hemp [27] and peaches [2,3]; however, it has never been described for perilla inflorescences. As reported by Pourcel et al., polyphenolic compounds can be oxidized within plants due to the increased activity of polyphenol oxidases and peroxidases during seed and plant development [28]. This oxidation process provides physicochemical protection to plant tissues against various stresses. These differences in the perilla seeds and flowers according to variety and growth period are believed to influence their antioxidant properties, PM2.5-induced respiratory protection, and anti-inflammatory and inhibitory effects on mucus hypersecretion and fibrosis in RPMI 2650 cells.

### 2.2. Individual Phenolic Acids and Flavones

Caffeic acid and RA are phenolic compounds with diverse medicinal applications, notably present in the Lamiaceae family, including perilla [29]. These compounds initiate steps in the biosynthesis of useful secondary metabolites. We analyzed CA and RA using UHPLC (Ultimate 3000, Thermo Scientific, Waltham, MA, USA). Figure 2, Appendix A shows the phenolic acid contents of perilla seeds and flowers by variety and growth period. CA and RA were generally higher in perilla flowers (198.23–421.72 μg/g, 2028.45–3930.41 μg/g) than in seeds (97.95–281.01 μg/g, 743.75–3129.10 μg/g) across all varieties and growth periods. The contents of CA and RA increased till 20 days post-flowering (CA: 305.09–421.72 μg/g, RA: 2920.77–3930.41 μg/g) but showed a decreasing trend from 30 days onward, with CA reducing to 198.23–270.38 μg/g and RA to 2028.45–2807.33 μg/g by the final measurement. The biosynthesis of RA involves a complex enzymatic shikimate pathway, which has been extensively studied during plant maturation [29]. This pathway is a well-known precursor pathway for the biosynthesis of phenolic compounds, such as RA. In perilla, the biosynthesis of RA involves several enzymatic steps, including hydroxyphenylpyruvate reductase, as identified by Lu et al. [30], which is part of the RA biosynthesis pathway and is expressed in various tissues of perilla [30]. The role of this pathway in producing precursors for RA synthesis underscores its importance in the metabolic processes of perilla and other Labiatae plants [31]. Based on the shikimate pathway identified in previous studies, we predicted the metabolic pathway of the major phenolic acids (CA and RA) in perilla according to the growth period, as shown in Figure 3. The increased accumulation of CA and RA in the early stages (20 d) of maturation after flowering could be due to the activation of various compounds, including hydroxyphenylpyruvate reductase, which is involved in phenolic acid precursor synthesis. Furthermore, since RA is a large proportion of the overall secondary metabolites, these differences in the perilla seed and flower according to the growth period are believed to have the greatest impact on their physiological activities.

Flavones such as luteolin (LT), apigenin (AG), and their derivatives, which are present in the leaves, seeds, and aerial parts, are the active constituents of perilla. Luteolin and apigenin are key bioactive constituents responsible for many biological activities, including RA and CA [12]. The LT and AG content of perilla seeds and flowers, analyzed by variety and growth period, are presented in Figure 2, Appendix A. LT and AG also tended to be higher in perilla flower (169.31–980.68 μg/g and 112.85–533.40 μg/g) than in perilla seed (97.93–783.89 μg/g and 63.46–253.86 μg/g) across all varieties and growth periods.

The trends exhibited by LT and AG were different from those shown by TPC, TFC, CA, and RA. LT and AG showed more variation between different varieties than between growth periods. The highest LT and AG content in perilla flowers was found in the M90 variety (769.71–980.68 μg/g for LT and 482.32–533.40 μg/g for AG), whereas the lowest content was found in the Anyu variety (169.31–218.81 μg/g for LT and 482.01–533.40 μg/g for AG). Although the variations in LT and AG with maturation were less significant compared to other secondary metabolites, we identified specific trends. LT accumulation steadily increased with maturation, reaching 980.68 μg/g at 40 days post-flowering, compared to 769.71 μg/g at 10 days. AG content ranged from 112.85 to 482.32 μg/g at 10 days after flowering, increased to 533.40 μg/g by 30 days, and then decreased to 482.01 μg/g. These differences in trends between LT and AG may be due to the conversion of AG to LT. Figure 3 represents the metabolic pathways of these major flavones (LT and AG) according to the growth period. Phenolic acid (caffeic acid and rosmarinic acid) and flavonoid (apigenin and luteolin) metabolic pathways of perilla change according to the maturation stage.

The first committed reaction in the flavonoid biosynthesis pathway is the condensation of one p-coumaroyl-CoA molecule with three malonyl-CoA molecules, catalyzed by chalcone synthase (CHS), which forms chalcones [32]. Chalcones are precursors for the biosynthesis of a range of flavonoids, including flavanones, flavones, isoflavones, flavonols, anthocyanins, and proanthocyanins [32]. They are unstable compounds that do not accumulate in plants [33]. The reaction catalyzed by chalcone isomerase converts chalcone into flavanones, such as naringenin, apigenin, eriodictyol, and luteolin [34]. The conversion of apigenin to luteolin might occur upon hydroxylation via eriodictyol, catalyzed by flavonoid hydroxylase (F3’H) [35]. According to a study by Zuk et al. [35] on temporal biosynthesis of flavone constituents in flax growth stages, at the flowering stage (here shown as weeks 9–11), the level of flavone was approximately five times higher than at the vegetative stage (weeks 2–8) and approximately 70% higher than at maturity. They reported that the fluctuation in apigenin and luteolin glucoside levels upon plant development correlated with the activity of F3′H.

### 2.3. Antioxidant Activities

Phenolic compounds such as phenolic acids and flavones are known for their antioxidant activities [36]. Therefore, assessing how changes in the content of these compounds resulting from plant development affect the antioxidant potential of plants is crucial [35]. The antioxidant activities of extract residues containing phenolic compounds, such as phenolic acid and flavones, were analyzed. The 2,2-azinobis (3-ethyl benzothiazoline)-6-sulfonic acid (ABTS) and 1,1-diphenyl-2-picrylhydrazyl (DPPH) radical scavenging activities of perilla seeds and flowers according to variety and growth period are shown in Figure 2, Appendix A. The ABTS radical scavenging activity in perilla seeds ranged from 11.03 to 16.03 mg TE/g ER, showing minimal variation across growth stages. In contrast, perilla flowers showed higher activity, ranging from 12.91 to 55.54 mg TE/g ER, with significant changes depending on maturation. Similarly, the DPPH radical scavenging activity was higher in perilla flowers (5.85–40.57 mg TE/g ER) than in seeds (5.99–11.41 mg TE/g ER), also showing a marked variation across maturation stages. Both ABTS and DPPH activities peaked at 20 days post-flowering and decreased significantly after 30 days. The main factor underlying the antioxidant qualities of perilla is the abundance of phenolic acids, flavones, and other polyphenolic chemicals, which differ depending on the plant part, variety, and growing environment. This difference in antioxidant activity between plant parts (perilla seeds and flowers) could be due to phenolic compounds being more abundant in flower buds and seed coats than in the seeds. As the proportion of seeds increased with plant maturation after flowering, the extract contained a higher sugar proportion, carbohydrates, and soluble proteins. In particular, the variation in the antioxidant activity of perilla seeds and flowers at different growth periods in this study was likely the result of increased biosynthesis and accumulation of phenolic acids such as CA and RA. Rosmarinic acid is a natural polyphenol with significant antioxidant properties, as demonstrated across various studies [37]. RA’s radical-scavenging efficacy of RA is further supported by computational studies, which show that RA exhibits strong scavenging properties for hydroxyl (HO·) and hydroperoxyl (HOO·) radicals in physiological environments, with rate constants significantly higher than those of standard antioxidants such as Trolox [38]. The proportion of RA to the total of the major individual compounds (CA, RA, RT, AG) was approximately 62–89%, which may have contributed the most to the antioxidant activity. Therefore, these variations in antioxidant properties, as well as RA in perilla seeds and flowers, according to the varieties and growth periods in the present study, are anticipated to affect physiological properties, including PM2.5-induced respiratory enhancement effects in RPMI 2650 cells.

### 2.4. Screening of Respiratory Improvement Activities

The respiratory improvement activities of perilla seeds and flowers were evaluated to investigate the effects of secondary metabolite variations according to different varieties and growth periods. This was done by analyzing their PM2.5-induced respiratory protective, anti-inflammatory, and inhibitory effects on mucus hypersecretion and fibrosis in RPMI 2650 cells. Treatment with PM2.5 (25–400 μg/mL) resulted in a decrease in cell viability and increased secretion of inflammatory cytokines (NO, TNF-α, IL-6, and IL-1β), the mucus-related biomarker (MUC5AC), and the fibrosis-related biomarker (MMP-9). Specifically, the application of PM2.5 (100 μg/mL) to stimulate RPMI2650 cells resulted in a 50% reduction in cell viability (Appendix A) and a notable increase in these biomarkers, establishing a cellular model for screening the respiratory disease enhancement effects of perilla seeds and flowers across various varieties and growth periods.

Sinonasal and nasal epithelial cells form a barrier that acts as a crucial defense mechanism against airborne pollutants. Elevated PM2.5 levels may cause oxidative stress, change mitochondrial metabolism, and damage intercellular connections, all of which might upset the nasal epithelium’s structure [39]. Nasal epithelial cells undergo inflammatory alterations after exposure to PM2.5 [40]. In this study, three biomarkers (cell viability, NO, and MUC5AC concentration) were evaluated for PM2.5-induced respiratory disease enhancement effects in the positive control (thymoquinone, 2.5 ppm), perilla seeds, and flowers of different varieties (DY, AY, M78, M90, and YCPL243) and growth periods (10 d, 20 d, 30 d, and 40 days after the flowering stage). Preliminary experiments were performed to determine the toxic concentrations of PM2.5 and FREs in RPMI 2650 cells using the 3-[4,5-dimethylthiazol-2-yl]-2,5-diphenyl-tetrazolium bromide (MTT) assay (Figure 4A). Although the exposure of RPMI 2650 cells to PM2.5 (100 μg/mL) resulted in the reduction of 46.48% cytotoxicity compared with the control cells, treatment with FREs did not exhibit cytotoxicity at concentrations <100 μg/mL for all perilla varieties and growth periods.

Next, we analyzed the cytoprotective effects and NO and MUC5AC inhibition effects of FREs in perilla seeds and flowers based on the variety and growth periods against PM2.5-induced toxic damage, which demonstrated distinct differences among the five varieties and four growth periods (Figure 4B and Figure 5A,B). Cell viability was decreased to 49.92% of control cells through PM2.5 inductions but increased from 46.16 to 74.31% with treatment using FREs of perilla seeds, and a higher increase from 63.89 to 99.59% in the case of perilla flowers. Cell viability of positive control was 68.86% after treatment with thymoquinone (25 ppm). The changes in the cell viability of perilla seed- and flower-derived FREs at different growth periods were similar to the results for TPC, TFC, RA, CA amounts, and antioxidant activity. Therefore, cell viability of FRE from perilla flower harvested 10 days after flowering stages ranged from 80.78 to 90.98%, but slightly increased from 88.67 to 99.58% on the 20 d, and showed a decreasing trend from 30 d, and finally significantly reduced from 63.89 to 73.59%. The concentration of NO secreted by nasal cells, a major biomarker of inflammation, increased by PM2.5 inductions from 7.45 to 44.39 μM, while it decreased to 25.33 μM by the positive control (thymoquinon, 25 ppm) treatment and decreased from 0.41 to 44.49 μM and 17.94 to 41.87 μM by FREs from perilla seeds and flowers, respectively. Further, the concentration of MUC5AC secreted by nasal cells, a major biomarker of mucus hypersecretion, increased by PM2.5 inductions from 10.25 to 66.56 ng/mL, while it decreased to 21.00 ng/mL by the positive control (thymoquinon, 25 ppm) treatment, and slightly decreased from 57.95 to 68.11 ng/mL and 46.10 to 57.29 ng/mL by FREs from perilla seeds and flowers, respectively.

Among the 40 extracts prepared from five varieties and four growth periods of perilla seeds and flowers, FRE from perilla flowers of Anyu cultivar harvested on the 20th day after the flowering stages exhibited significant respiratory protective effects against fine dust stimulation while effectively inhibiting NO (major inflammation indicator; Figure 5) and MU5AC. Thymoquinone found in *Nigella sativa*, a plant utilized as a positive control in this study, reduces inflammation, oxidative stress, apoptosis, and autophagy in PM2.5-induced lung damage by decreasing inflammatory response markers (IL-1β, IL-6, and TNF-α), as well as oxidative stress [41].

The anti-allergic and anti-inflammatory effects of PFD-derived compounds have been reported in numerous in vitro and in vivo models [42]. Specifically, PLEs demonstrated significant anti-inflammatory effects on non-tumorigenic human bronchial epithelial cells (BEAS-2B) [43]. In dermatophagoides pteronyssinus 2 (DP2)-stimulated BEAS-2B cells, treatment with PLEs drastically reduced the mRNA expression and protein levels of pro-allergic and pro-inflammatory cytokines via blocking P38/c-Jun N-terminal kinases (JNK) and NF-kB activation.

While compounds from perilla seeds and leaves are recognized as functional materials for reducing inflammation and improving respiratory disease in the nasal passages, bronchi, and lungs, research on the protective effects of perilla flowers, which are rich in secondary metabolites, is limited. Furthermore, no studies have determined the growth periods when the accumulation of phenolic compounds is the highest, and their effects on respiratory protection have been investigated. Therefore, by screening experiments on optimal varieties and growth periods, we identified perilla seeds and flowers harvested 20 days after the flowering stage with distinct enhancing effects on respiratory health. We studied their PM2.5-induced respiratory protective, anti-inflammatory, and inhibitory effects on mucus hypersecretion and fibrosis in RPMI 2650 cells.

### 2.5. Correlation Analysis and Principal Component Analysis (PCA)

The results of the correlation analysis among TPC, TFC, individual phenolic compounds (CA, RA, RT, AG), antioxidant activities (ABTS and DPPH radical scavenging activity), cell viability, NO, and MUC5A of perilla seeds and flowers, based on variety and growth period, are shown in Figure 6A. The antioxidant compounds and activities, such as TPC, TFC, ABTS, and DPPH radical scavenging activities, exhibited significant positive correlations with cell viability and negative correlations with NO and MUC5AC concentrations and biomarkers of inflammation and mucus hypersecretion. Among the individual phenolic compounds, RA showed the highest positive correlation with cell viability, whereas RT and AG exhibited slight correlations with significant factors affecting the respiratory enhancement effect. Conversely, policosanol, the main fat-soluble component of perilla seeds, negatively correlated with both antioxidant activity and respiratory enhancement.

Principal component analysis (PCA) is commonly employed to synthetically analyze clustering trends in multidimensional data. It was employed to assess the clustering and variations in functional compounds and significant factors affecting the antioxidant and respiratory protective properties of perilla seeds and flowers according to the variety and growth period (Figure 6B). PC1 and PC2 accounted for 92.9% and 6.8% of the variance, respectively. Cluster analysis confirmed that the metabolites and activities of perilla seeds and flowers were distinguishable across all varieties and growth periods. Based on correlation and PCA results, RA was the individual compound with the most significant effect on the respiratory health benefits of perilla flower extract in this study. RA has been investigated for its potential therapeutic effects on respiratory conditions, particularly through its antioxidant, anti-inflammatory, and anti-apoptotic properties [13]. Although no studies have reported the respiratory health benefits of the early maturation of perilla flowers, we postulate that the respiratory health benefits of RA have been reported in previous studies. In this study, perilla flowers 20 days after the flowering stage were the most effective in RA biosynthesis. The substantial accumulation of RA is believed to have enhanced antioxidant activity, improved cell viability against PM2.5, and inhibited NO production, a biomarker of inflammation, and MUC5AC, a biomarker of mucus hypersecretion.

### 2.6. PM2.5-Induced Respiratory Protection Effects in RPMI 2650 Cells of Optimal Perilla Extracts

PM2.5, a significant environmental pollutant, is known to induce oxidative stress, which is a critical mechanism underlying its adverse health effects [44]. Oxidative stress occurs when there is an imbalance between reactive oxygen species (ROS) production and the respiratory ability to detoxify these reactive intermediates or repair the resulting damage. PM2.5 exposure is associated with increased oxidative stress markers, such as malondialdehyde (MDA), Nε-(hexanoyl)-lysine (HEL), and 8-hydroxy-2-deoxyguanosine (8-OHdG), which are indicative of lipid peroxidation and DNA damage [45]. Chronic intranasal exposure to fine dust can cause lung inflammation, mucosal hypersecretion, and pulmonary fibrosis. This results in oxidative stress, which is associated with the TGFβ1-PI3K/Akt, TGFβ1-NOX, and TGFβ1-NLRP3 pathways, impacting mucus production and fibrosis [46].

In this study, the respiratory protective and inhibitory effects of PFEs from perilla seeds and flowers on inflammation, mucus hypersecretion, and fibrosis under optimal conditions (Anyu, Day 20) were identified in PM2.5-induced damage in RPMI 2650 cells. First, cell viability, ROS production, and MDA concentrations were analyzed to evaluate respiratory protective effects (Figure 7A–C). RPMI 2650 cells were treated with three-point concentrations of the samples (50, 100, 200 μg/mL) for 24 h with PM2.5 (100 μg/mL). These results demonstrate that treatment with PM2.5 considerably reduced cell viability by 47.88% compared to that in control cells. However, PFE treatment from perilla seeds and flowers substantially increased cell viability in a dose-dependent manner. Specifically, the protective effects against PM2.5-induced RPMI 2650 cell damage were excellent in PFEs of perilla flowers compared to perilla seeds, which increased cell viability by 100.80%. PM2.5, treatment considerably stimulated oxidative cellular stress and damage, enhancing ROS release in RPMI 2650 cells by 234.18% compared to those without PM2.5 (control cells). However, treatment with PFEs from perilla flowers compared to perilla seed substantially reduced PM2.5-induced intracellular ROS levels in RPMI 2650 in a dose-dependent manner. Specifically, PFEs from perilla flower reduced ROS levels to 125.17% at a concentration of 200 μg/mL Subsequently, the PM2.5-induced intracellular MDA concentration in RPMI 2650 cells was analyzed as a lipid peroxidation index.

Oxidative stress-induced PM2.5 in RPMI2.50 cells stimulated a significant increase in MDA concentration. However, pretreatment with PFEs (200 μg/mL) counteracted this increase, with PLEs being the most effective. This study demonstrated that PFEs, specifically perilla flowers harvested at an early stage (20 d) after flowering, provided respiratory protective effects distinct from those of common perilla leaves. These extracts effectively enhanced cell viability in the presence of PM2.5 reduced ROS levels and suppressed lipid peroxidation.

To evaluate the inhibitory effects of PFEs from optimally harvested perilla seeds and flowers (Anyu, Day 20) on inflammation, mucus hypersecretion, and fibrosis, concentrations of NO, TNF-α, IL-6, MUC5AC, and MMP-9 secreted by PM2.5-induced RPMI 2650 cells were analyzed. PM2.5 exposure significantly increased the production of these markers: NO (41.22 μM), TNF-α (153.25 ng/mL), IL-6 (64.17 ng/mL), MUC5AC (75.44 ng/mL), and MMP-9 (79.37 pg/mL), compared to untreated cells (6.33 μM, 11.40 ng/mL, 3.60 ng/mL, 9.49 ng/mL, and 50.60 pg/mL, respectively). Perilla seed and flower extracts considerably reduced these levels in a dose-dependent manner, with perilla flower extracts proving more effective, reducing the concentration of NO, TNF-α, IL-6, MUC5AC, and MMP-9 by 10.33 μM, 48.82 ng/mL, 19.07 ng/mL, 35.46 ng/mL, and 42.25 pg/mL, respectively.

MUC5AC, a major mucin in the respiratory system, plays a crucial role in mucociliary clearance by forming a protective hydrogel barrier against pathogens. However, dysregulation of this gene contributes to various respiratory diseases. MUC5AC expression is significantly increased in patients with asthma, specifically in those with cold airway hyper-responsiveness, resulting in an imbalance in mucin production that exacerbates disease severity [47]. Additionally, MMPs play a crucial role in the pathogenesis and progression of various respiratory diseases, specifically fibrosis. They are zinc-dependent endopeptidases that degrade extracellular matrix components, which are crucial for tissue remodeling and repair. However, an imbalance in MMP activity and their inhibitors can result in pathological tissue destruction and fibrosis, as observed in conditions, such as idiopathic pulmonary fibrosis, chronic pulmonary tuberculosis, and other interstitial lung diseases [48].

Understanding the regulatory mechanisms underlying mucus secretion and MMPs is crucial for preventing respiratory diseases. According to Pintha et al. [49], RA mitigates oxidative stress, inflammation, and cancer metastasis in A549 lung cancer cells. This is achieved through the suppression of key signaling pathways, such as c-Jun, p-65-NF-κB, and Akt, which are activated by PM exposure and contribute to the production of ROS and pro-inflammatory cytokines like IL-6, IL-8, and TNF-α. RA also reduced the activity of MMP-9, which is involved in cell migration and invasion, suggesting its potential as a therapeutic agent for lung inflammation and cancer metastasis prevention due to PM exposure. Perilla flower extracts at the early maturation stage (20 d) effectively inhibited mucus hypersecretion and fibrosis in PM2.5-induced RPMI 2650 cells, demonstrating their respiratory protective and anti-inflammatory effects. Although research on the respiratory health benefits of early-matured perilla flowers is limited, we postulate that these benefits may be attributed to RA, as reported in previous studies. Therefore, we investigated how variations in secondary metabolites contribute to the health benefits of perilla seeds and flowers, including their respiratory protective effects. Furthermore, we confirmed that the early maturation of perilla flowers after flowering specifically protected the respiratory system by reducing oxidative stress, inflammation, mucus hypersecretion, and fibrosis.

## 3. Materials and Methods

### 3.1. Materials

Five types of perilla seeds, including two Korean cultivars (*P. frutescens*, cv. Dayu, cv. Anyu), two elite lines (Milyang 78 and Milyang 90), and one genetic resource (YCPL243), were grown at the National Institute of Crop Science in Miryang, South Korea, during the 2022 growing season. Five Perilla cultivars with distinct characteristics were evaluated for edible seed production: ‘Dayu’ is characterized by high oil content, white flower color, and brown grain color, while ‘Anyu’ exhibits early maturity, high oil content, purple flower color, and brown grain color. Additionally, Milyang 78 and Milyang 90 were identified as elite lines with high rosmarinic acid and oil content, and YCPL243, as a genetic resource, was noted for its high rosmarinic acid content and excellent antioxidant activity. To investigate changes in secondary metabolites and respiratory health benefits according to growth periods and plant parts, perilla seeds and flowers were harvested and freeze-dried at 10, 20, 30, and 40 days after flowering. The flowering dates for each variety were Dayu on 5 September, Anyu on 25 August, Milyang 78 on 2 September, Milyang 90 on 30 August, and YCPL243 on 6 September. The samples were harvested 10–40 days after each flowering time.

### 3.2. Preparation of Flavonoid-Rich Extracts

FRE from perilla seeds and flowers of various varieties and growth periods were prepared using an optimized method [50]. Briefly, 2 g of powdered samples were extracted twice with 40 mL of 60% ethanol at 80 °C for 2 h using a shaking water bath. The extracts were then filtered and concentrated using a rotary evaporator under vacuum, freeze-dried, and stored at −20 °C in an ultralow temperature freezer.

### 3.3. Functional Compounds Analysis in Perilla Seed and Flower of Different Growth Periods

The total polyphenol and flavonoid levels were measured according to the method described by Woo et al. [51]. Total polyphenol levels were measured using the Folin–Ciocalteu method, where 10 μL of standards or extracts were mixed with 200 μL of 2% (*w*/*v*) sodium carbonate solution and 10 μL of 50% (*v*/*v*) Folin–Ciocalteu reagent (Sigma-Aldrich, St. Louis, MO, USA). The mixtures were incubated for 30 min at room temperature, and absorption was measured at 750 nm. The results were expressed as milligrams of gallic acid (Sigma-Aldrich) equivalents per gram of extract residue (GAE/g ER). To measure total flavonoid levels, we mixed 50 μL of standards or extracts were mixed with 200 μL water and 15 μL of 5% (*w*/*v*) NaNO_2_. After 5 min, 30 μL AlCl_3_·6H_2_O (10%, *w*/*v*) was added and the incubation was continued for another 6 min. The reactions were terminated by adding 1 M NaOH (100 μL), and the absorbance at 510 nm was measured. The results are expressed as milligrams of gallic acid and catechin equivalents per gram of extract residue (mg GAE/g ER, mg CE/g ER). The individual phenolic compound compositions (RA, caffeic acid, and luteolin) of perilla seeds and flowers were determined by high-performance liquid chromatography (HPLC), as described by An et al. [52]. The poleosanol (Hexacosanol, Octacosanol, triacosanol) content of perilla seeds and flowers was analyzed using a 7890A GC system (Agilent Technologies, Santa Clara, CA, USA) equipped with a 5975C MSD (Agilent Technologies), as described by Park et al. [53].

### 3.4. Antioxidant Activities in Perilla Seed and Flower of Different Growth Periods

The radical-scavenging activities of 1,1-diphenyl-2-picrylhydrazyl (DPPH) and 2,2-azinobis (3-ethyl benzothiazoline)-6-sulfonic acid (ABTS) were measured according to methods reported by Woo et al. [51]. For DPPH activity, a 1000 μL aliquot of 0.2 mM DPPH (1,1-diphenyl-2-picrylhydrazyl; Sigma-Aldrich) methanolic solution was mixed with 50 μL of each sample, shaken vigorously, and left to stand for 30 min under low light. Absorbance was then measured at 515 nm. We generated ABTS cationic radicals by adding ABTS (Sigma-Aldrich) to a concentration of 7 mM in a 2.45 mM potassium persulfate solution and allowing the mixture to stand overnight in the dark at room temperature. This radical solution was diluted with methanol to achieve an absorbance of 1.4–1.5 at 735 nm (molar extinction coefficient, ε = 3.6 × 10^4^ mol^−1^ cm^−1^). For each test, 1000 μL of the diluted ABTS radical solution was added to 50 μL of each extract, a Trolox standard solution, or distilled water. After 30 min, absorbance was measured at 735 nm spectrophotometrically (Multiskan GO Microplate Spectrophotometer; Thermo Fisher Scientific, Waltham, MA, USA). Both radical scavenging activities were expressed as Trolox-equivalent antioxidant capacity and as mg TE/g perilla seed and flower extract residues (ER).

### 3.5. Screening Analysis for Improvement Effect of Respiratory Disease

The RPMI 2650 cell line, derived from squamous cell carcinoma of the nasal septum, was obtained from the Korean Cell Line Bank (Seoul, Republic of Korea). The RPMI 2650 cells, NCI-H292 cells, and A549 cells were maintained in RPMI 1640 supplemented with 10% fetal bovine serum, 100 U/mL penicillin, and 50 μg/mL streptomycin in a 5% CO_2_ incubator at 37 °C. Cells were sub-cultured in 0.05% trypsin-EDTA and phosphate-buffered saline (PBS).

The toxicity of RPMI 2650, NCI-H292, and A549 cells to fine particulate matter (PM2.5; 25, 50, 100, 200, 400 μg/mL), FRE (25, 50, 100, 200, 400 μg/mL) from perilla seed and flower of control cultivar (Dayu), and FRE (100 μg/mL) from five perilla seed and flower varieties according to growth periods (10, 20, 30, 40 d) was analyzed using an MTT (3-[4,5-dimethylthiazol-2-yl]-2,5 diphenyl tetrazolium bromide) assay [54]. After determining the optimal cell lines (RPMI-2650), fine particulate matter concentration (100 μg/mL), and FRE (100 μg/mL) from perilla seed and flower of control cultivar (Dayu), we evaluated respiratory protective effects against PM2.5 through mass screening system for cytoprotective effect (cell viability, %), inflammatory response (nitric oxide concentration, μM), and mucus production (MUC5AC, ng/mL). RPMI 2650 cells were seeded in a 96-well plate at a density of 5 × 10^4^ cells/well to determine cytoprotective effects against fine particulate matter (PM2.5 like; 100 μg/mL). After 24h, FRE (100 μg/mL) from perilla seed and flower with fine particulate matter (PM2.5 like; 100 μg/mL) was administered to the cells. After 24 h incubation, cell viability was determined using the MTT assay. Simultaneously, nitric oxide (NO) levels and Human Mucin 5AC (MUC5AC) concentrations in the supernatants were measured using a Griess Reagent System (Promega, Madison, WI, USA) and a Human Mucin 5AC ELISA kit (Abcam, Cambridge, UK).

### 3.6. Respiratory Protective Effect in PM2.5-Induced RPMI 2650 Cells of Optimized Perilla Extracts

Human nasal RPMI 2650 cells were seeded in a 96-well plate at a density of 2 × 10^4^ cells/well to determine their cytoprotective effects against PM2.5 (100 μg/mL) [55]. After 24 h, optimized perilla extract (OPE; 50, 100, 200 μg/mL) from seeds and flowers harvested on day 20 after flowering time of the selected cultivar (Anyu) was administered with PM2.5 (100 μg/mL). After incubation for 24 h, cell viability was determined using an MTT assay as previously described. Human nasal RPMI 2650 cells cultured in a 24-well plate at 1 × 10^6^ cells/well were used for intracellular ROS detection, lipid peroxidation, and antioxidant activity. Intracellular reactive oxygen species (ROS) levels were quantified using a DCFH-DA fluorescent probe, as previously described [56]. Fluorescence intensity, corresponding to intracellular ROS generation, was measured using a fluorescence spectrophotometer for 2 h at an excitation wavelength of 485 nm and an emission wavelength of 530 nm. Cells were harvested and lysed using radioimmunoprecipitation assay buffer (Sigma, St. Louis, MO, USA) to determine lipid peroxidation and antioxidant enzyme activity. Lysates were centrifuged at 10,000× *g* for 10 min at 4 °C, and the supernatants were analyzed for protein, lipid peroxidation, and antioxidant enzyme activity [57].

### 3.7. Anti-Inflammatory Effect in PM2.5-Induced RPMI 2650 Cells of Optimized Perilla Extracts

Human nasal RPMI 2650 cells, cultured in 24-well plates at a density of 1 × 10^6^ cells per well, were used for nitric oxide (NO) and cytokine such as TNF-α, IL-6, IL-1β production analysis [58]. After 24 h, optimized perilla extract (OPE; 50, 100, 200 μg/mL) from seeds and flowers harvested on day 20 after flowering time of the selected cultivar (Anyu) was administered to the cells along with suspended fine particulate matter (PM2.5, 100 μg/mL). After incubation for 24 h, the concentrations of nitric oxide (NO) and the cytokines such as TNF-α, IL-6, and IL-1β in cell-free supernatants were determined using a Griess Reagent System (Promega, Madison, WI, USA) and ELISA kits for TNF-α, IL-6, and IL-1β (R&D Systems, Abingdon, UK), according to the manufacturer’s instructions.

### 3.8. Inhibition Effect of Mucus Hypersecretion and Fibrosis in PM2.5-Induced RPMI 2650 Cells of Selected Resources

Human nasal RPMI 2650 cells, cultured in 12-well plates at 2 × 10^6^ cells per well, were used to evaluate mucus hypersecretion and fibrosis inhibition against fine particulate matter (PM2.5) [59]. Mucus hypersecretion was assessed by analyzing intracellular Mucin-5AC (MUC5AC) and fibrosis was assessed by analyzing matrix metalloproteinase-9 (MMP-9) expression. After 24 h, optimized perilla extract (OPE; 50, 100, 200 μg/mL) from seeds and flower harvested on day 20 after flowering time of the selected cultivar (Anyu), was administered to the cells were added with suspended fine particulate matter (PM2.5, 100 μg/mL). After incubation for 24 h, the cells were harvested and lysed using radioimmunoprecipitation assay buffer (RIPA, Sigma, St. Louis, MO, USA) to determine MUC5AC and MMP-9 concentrations. Lysates were centrifuged at 10,000× *g* for 10 min at 4 °C, and the supernatants were used for the analysis to determine MUC5AC and MMP-9 concentrations using an ELISA kit (R&D Systems, Abingdon, UK), according to the manufacturer’s instructions.

### 3.9. Statistical Analysis

All data are expressed as means ± standard deviations. Significant differences among treatments were determined by a one-way analysis of variance (ANOVA) using Duncan’s multiple range test in SAS ver. 9.2 software (SAS Institute, Cary, NC, USA). The significance level was set at *p* < 0.05. Correlation and principal component analysis (PCA) were applied through loading plots, score plots, and heatmaps to evaluate the results using the MetaboAnalyst 5.0, a platform for metabolomics data analysis (one factor) (https://www.metaboanalyst.ca).

## 4. Conclusions

This study revealed the variation in phenolic acids, flavonoids, and polycosanols (Appendix A) in perilla flowers and seeds at different post-flowering developmental stages and assessed their protective effects against PM2.5-induced respiratory damage, anti-inflammatory responses, and inhibition of mucus hypersecretion and fibrosis in RPMI 2650 cells. CA and RA showed increased accumulation in the early maturation stage (20 days post-flowering), whereas AG and LT continued to increase until later stages (30 and 40 days, respectively). These variations in metabolite accumulation impacted the antioxidant and respiratory-enhancing effects. Notably, flower extracts harvested 20 days after flowering exhibited strong respiratory protective effects against fine dust exposure and significantly inhibited NO and MU5AC production. Correlation analysis identified RA as the key compound contributing to the observed respiratory health benefits. These extracts also protected the respiratory system by reducing oxidative stress, inflammation, mucus hypersecretion, and fibrosis. In conclusion, perilla flower extracts, particularly from the 20-day post-flowering stage, demonstrated strong protective effects on respiratory health by reducing oxidative stress, inflammation, mucus hypersecretion, and fibrosis, largely due to the accumulation of RA and other key functional components.

## Figures and Tables

**Figure 1 ijms-25-12119-f001:**
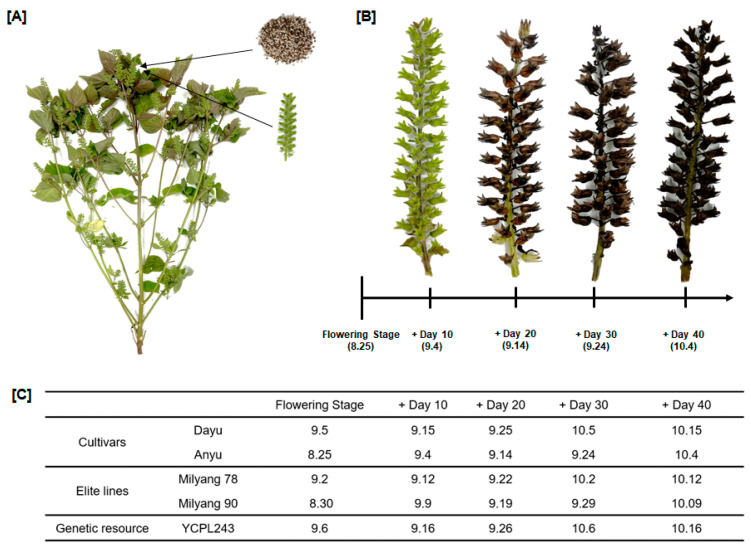
Flowering stage and growth periods of five varieties of perilla seeds and flowers. (**A**) Perilla frutescens plant. (**B**) Perilla flower of Anyu cultivar at 10, 20, 30, and 40 days after flowering stage. (**C**) Flowering stage and harvesting times according to 5 different varieties.

**Figure 2 ijms-25-12119-f002:**
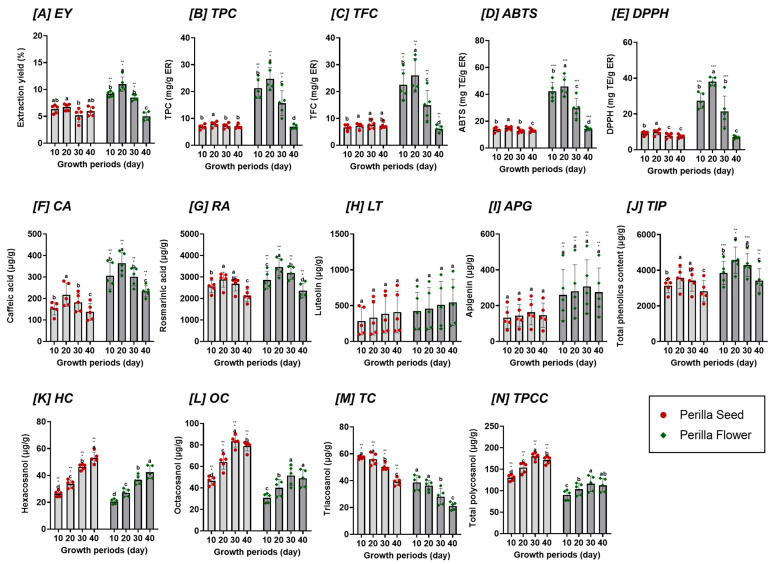
Distribution of extraction yield [EY], antioxidant compounds (TPC, TFC), antioxidant activities (ABTS and DPPH radical scavenging activity), total individual phenolics [TIP], caffeic acid [CA], rosmarinic acid [RA], Luteolin [LT], Apigenin [APG], total policosanol [TPCC], Hexacosanol [HC], Octacosanol [OC], and Triacosanol [TC] in perilla seeds and flowers, based on varieties and growth periods. Values are the mean ± SD of 3 replicates. Different small letters in the same items indicate a significant difference (*p* < 0.05) among different growth periods. *** *p* < 0.001, ** *p* < 0.01 and * *p* < 0.05 represent significant differences between perilla seed and perilla flower.

**Figure 3 ijms-25-12119-f003:**
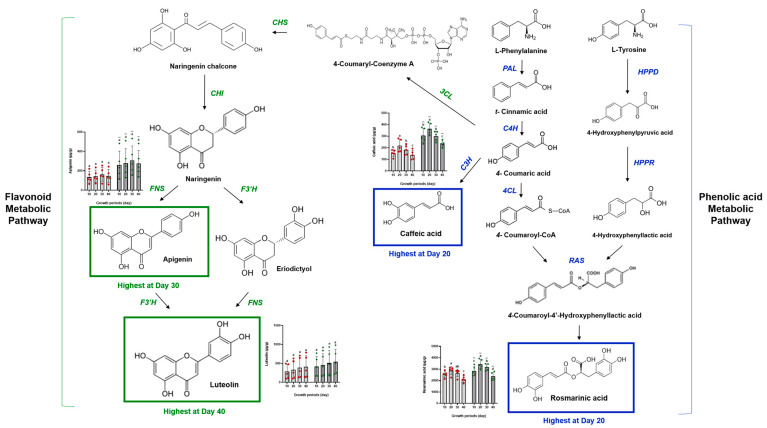
Phenolic acid (caffeic acid and rosmarinic acid) and flavonoid (apigenin and luteolin) metabolic pathway of perilla according to maturation stage. Values are the mean ± SD of 3 replicates. Different small letters in the same items indicate a significant difference (*p* < 0.05) among different growth periods. *** *p* < 0.001 represent significant differences between perilla seed and perilla flower.

**Figure 4 ijms-25-12119-f004:**
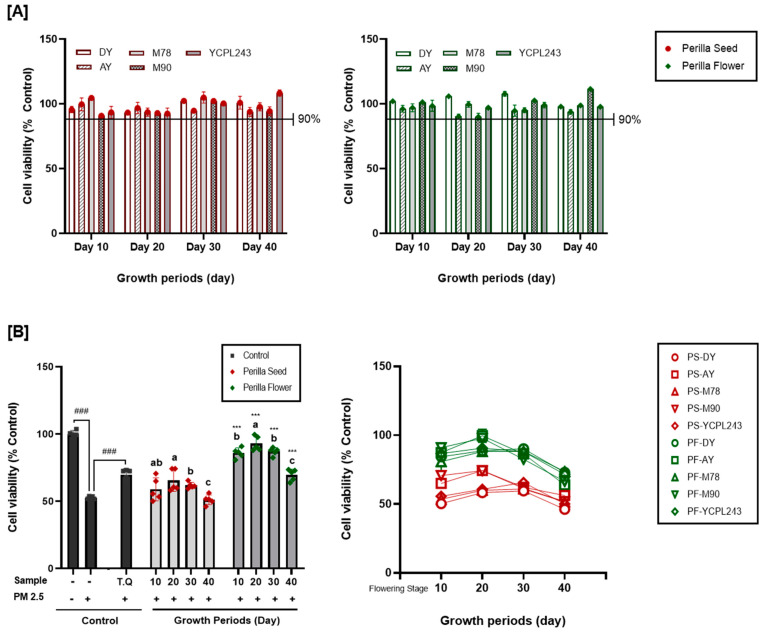
Respiratory disease enhancement effects of perilla seed and flower extracts with different varieties and growth periods in PM2.5-induced human nasal cell. (**A**) Cytotoxicity of extracts in human nasal (RPMI 2650) cell, (**B**) cell viability in PM2.5-exposed human nasal (RPMI 2650) cells. Values are the mean ± SD of 3 replicates. Different small letters in the same items indicate a significant difference (*p* < 0.05) among different growth periods. *** *p* < 0.001 represents significant difference between perilla seed and perilla flower. ^###^ *p* < 0.001 represents significant difference compared to PM2.5 treated control.

**Figure 5 ijms-25-12119-f005:**
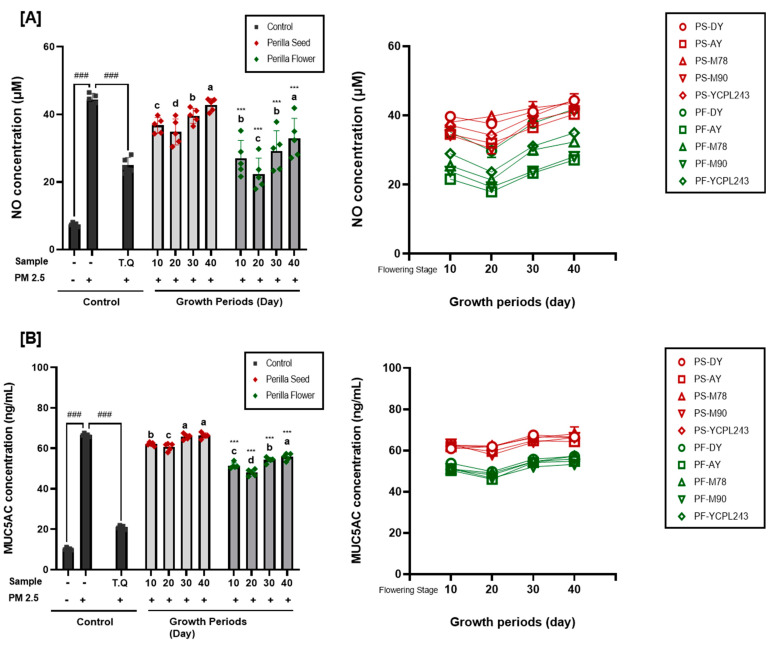
Respiratory disease enhancement effects of perilla seed and flower extracts with different varieties and growth periods in PM2.5-induced human nasal cell. (**A**) nitric oxide (NO), and (**B**) mucin 5AC (MUC5AC) secretion in PM2.5-exposed human nasal (RPMI 2650) cells. Values are the mean ± SD of 3 replicates. Different small letters in the same items indicate a significant difference (*p* < 0.05) among different growth periods. *** *p* < 0.001 represents significant difference between perilla seed and perilla flower. ^###^ *p* < 0.001 represents significant difference compared to PM2.5 treated control.

**Figure 6 ijms-25-12119-f006:**
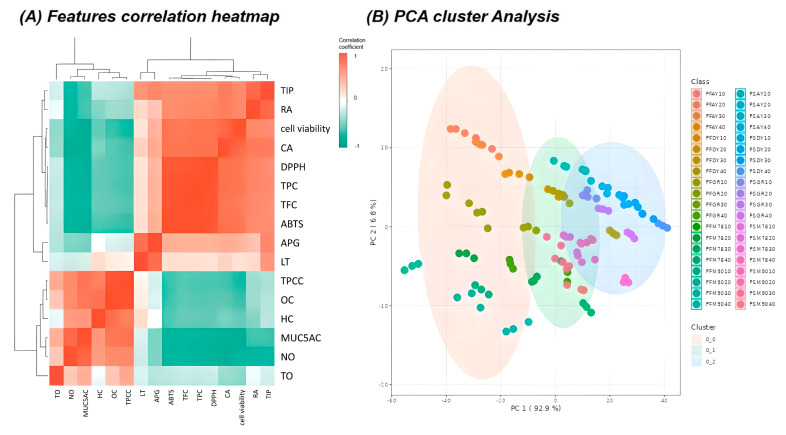
Correlation heat map analysis of the assigned antioxidant compounds, antioxidant activities, individual phenolic compounds, policosanol, and respiratory disease enhancement effects obtained from perilla (*Perilla frutescens* L.) seed and flower in various varieties and growth periods.

**Figure 7 ijms-25-12119-f007:**
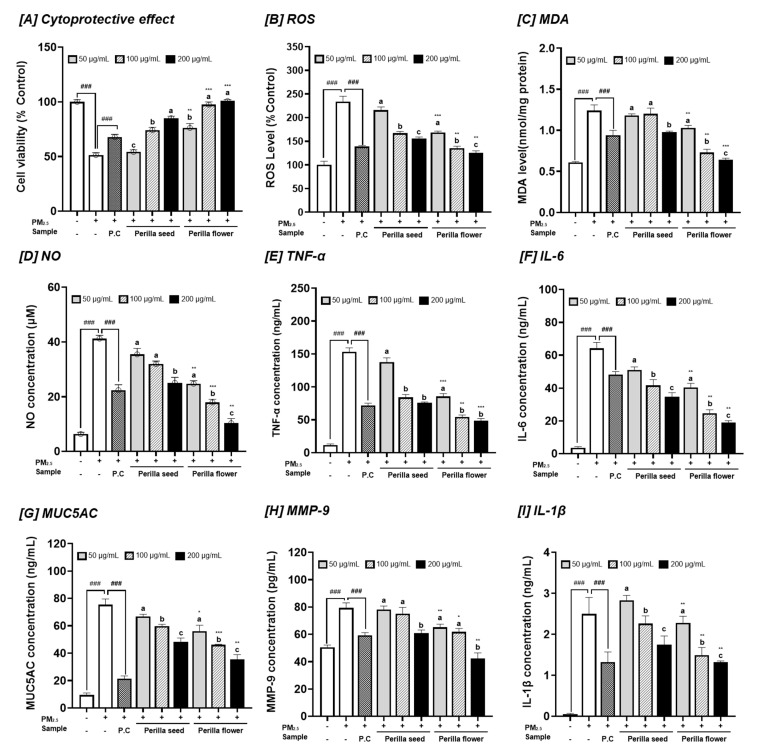
PM2.5-induced respiratory protective effect of selected cultivars (Anyu) at 20 days after flowering of perilla flowers in RPMI 2650 cells compared to perilla seed. Assessment of cytoprotective effect, reactive oxygen species (ROS) levels, lipid peroxidation of malonaldehyde (MDA; nmol mg^−1^ protein), nitric oxide (NO), tumor necrosis factor-alpha (TNF-α), interleukin (IL)-6, mucin 5AC (MUC5AC), and matrix metalloproteinase-9 (MMP-9) concentrations. Values are the mean ± SD of 3 replicates. Different small letters in the same items indicate a significant difference (*p* < 0.05) among different concentrations of extracts. *** *p* < 0.001, ** *p* < 0.01 and * *p* < 0.05 represent significant differences between perilla seed and perilla flower. ^###^ *p* < 0.001 represents significant difference compared to PM2.5 treated control.

## Data Availability

Data are contained within the article or Appendix A.

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
