# Peer review of "Influence of Secondary Metabolites According to Maturation of Perilla (Perilla frutescens) on Respiratory Protective Effect in Fine Particulate Matter (PM2.5)-Induced Human Nasal Cell"

_ijms, 2024, doi:10.3390/ijms252212119_

Round 1

Reviewer 1 Report

Comments and Suggestions for Authors

This manuscript by Kim et al. presents the secondary metabolite profiles of perilla seeds and flowers at different developmental stages and evaluates the respiratory protective capacity of their extracts. They identified 20-day post-flowering flowers as the most effective perilla organ for treating respiratory, inflammation, and oxidation disorders. In general, the study was well-designed, and the manuscript was well-written. My comments on the manuscript are below:

1.     The title is too long and needs to be revised. The following is a suggestion: “Influence of Developing Perilla Flower and Seed Secondary Metabolites on Respiratory Protective Effect in Fine Particulate Matter (PM2.5) Induced Human Nasal Cell.”

2.     Lines 27. Early stage? Please revise it.

3.     Line 42. Delete the full stop before reference 7.

4.     Are the morphological characteristics of flowers collected at 10, 20, and 30 days post-flowering different? What makes these flowers different? Provide photos if possible.

5.     In the abstract, you mentioned 10, 20, and 30 d, but 10, 20, 30, and 40 d in the method. Please revise.

6.     Line 223. “daylight hours are reduced from 11 to 14 hours or less?”

7.     Reduce the length of the “results and discussion.” Don’t over-discuss your results. Keep your findings easily readable. It would be more appreciated if the two sections were separated.

8.     Revise the conclusion.

9.     Revise figure legends. Provide a caption for sub-figures A, B, C, etc.

10.  Move Figure 3 to supplementary files.

11.  Improve the quality of Figure 7.

Comments on the Quality of English Language

Revise typographic errors throughout the manuscript.

Author Response

Point-by-point responses to the Editor’s and Reviewers’ comments

Reviewer Comments:

Reviewer 1

This manuscript by Kim et al. presents the secondary metabolite profiles of perilla seeds and flowers at different developmental stages and evaluates the respiratory protective capacity of their extracts. They identified 20-day post-flowering flowers as the most effective perilla organ for treating respiratory, inflammation, and oxidation disorders. In general, the study was well-designed, and the manuscript was well-written. My comments on the manuscript are below:

  1. The title is too long and needs to be revised. The following is a suggestion: “Influence of Developing Perilla Flower and Seed Secondary Metabolites on Respiratory Protective Effect in Fine Particulate Matter (PM2.5) Induced Human Nasal Cell.”

→ Thank you for your advice. Following your suggestion, I have shortened the title slightly

  1. Lines 27. Early stage? Please revise it.

→ Thank you for your comment. I have deleted the term “early stage” to avoid any potential confusion (Line 27).

  1. Line 42. Delete the full stop before reference 7.

→ Thank you for your comment. I have deleted the full stops before references 5 and 7(Line 42).

  1. Are the morphological characteristics of flowers collected at 10, 20, and 30 days post-flowering different? What makes these flowers different? Provide photos if possible.

→ Thank you for your comment. As per your advice, the morphological characteristics can vary depending on the maturity stage after flowering. Therefore, the morphological differences of the flower clusters at 10, 20, 30, and 40 days post-flowering can be observed in Figure 1(B).

  1. In the abstract, you mentioned 10, 20, and 30 d, but 10, 20, 30, and 40 d in the method. Please revise.

→ Thank you for your advice. I have harvested perilla flower clusters at 10, 20, 30, and 40 days post-flowering. Therefore, I have added the 40-day stage to the abstract (Line 19).

  1. Line 223. “daylight hours are reduced from 11 to 14 hours or less?”

→ Thank you for your advice. I have revised the sentence to avoid confusion. (Line 223)

  1. Reduce the length of the “results and discussion.” Don’t over-discuss your results. Keep your findings easily readable. It would be more appreciated if the two sections were separated.

→ Thank you for your comments. Since it is difficult to completely separate the results and discussion, I have followed your advice and revised some of the result descriptions to be more concise.

  1. Revise the conclusion.

→ Thank you for your comments. Although I am not entirely sure of the reviewer's intent regarding which part of the conclusion needs revision, I have supplemented and revised it overall to better encapsulate the findings of this study.

  1. Revise figure legends. Provide a caption for sub-figures A, B, C, etc.

→ Thank you for your comments. The sub-figures of Figure 1,5,6,7,8 are already captioned. Therefore, we have added captions for the sub-figures of Figure 2 and 8.

  1. Move Figure 3 to supplementary files.

→ Thank you for your comments. Figure 3 presents essential results as it allows for the comparison of differences between cultivars, unlike Figure 2. However, since the reviewer seems concerned about redundancy with Figure 2, I will move Figure 3 to the Supplementary Data section.

  1. Improve the quality of Figure 7.

→ Thank you for your comments. I have improved the quality of Figure 7.

Reviewer 2 Report

Comments and Suggestions for Authors

The authors study the role of different flowering stages of Perilla frutescens in inhibition of PM2.5 induced oxidative stress, inflammation, mucus production and fibrosis. Firstly, the authors need to pay extensive attention to the typing errors and components of the manuscripts. For example, legends are not matching with Figures. This is extremely confusing as readers. Please screen the manuscript for grammatical errors and typographical errors. The authors have performed extensive experiments, but the poor visibility of the figures, and usage of one commercial cell line to measure the limited readouts (ROS, NO, MDA, TNF-a, IL-6, MUC5AC and MMP-9) reduces the enthusiasm of the study.

·        Please increase the size and readability of the axes in the Figures 2,3 and 4, 7, 8. It is very hard to read the figures.

·        Please remove one of the two Figures 2 and 3 since they convey the same information.

·        Please show the results from the screening that was performed to determine the dosage of PM2.5, at least in supplemental.

·        Please correct/update Figure 5C. The first bar shows sample (-) and PM2.5(+) with almost 100% viability. But second bar also shows sample (-) and PM2.5(+) with almost 50% viability. How is that possible?

·        Please re write the legends properly. For example, the authors indicate write that Figure 5 B and C demonstrates changes in NO and MUC5AC, but Fig 5 only shows cell viability data. Similarly, the legends for Fig 5 and 6 are exactly same, but Fig 6 is NO and MUC5AC. As authors of scientific manuscripts, it is important to pay attention to what is written in the legends and if that corresponds to the Figures.

·        Please improve the font size and readability of Fig 7. I was not able to read it at all.  There is no point is putting figures if no one can read it.

·        In Fig 8, the authors state “mechanism” in line # 548 and 549. However, this is not a mechanism, it is pharmacological readouts. Please change the word “mechanism”.

·        Elaborate the reason for measuring NO and why is this data relevant to the study?

·        The authors state this flower can protect against fibrosis, and just indicate the levels of MMP-9. Firstly, the role of MMP-9 is unclear, as in whether it is pro or anti-fibrotic, making it a weak candidate for evaluation of fibrosis. I think the authors need to demonstrate additional fibrosis marker changes, like TIMPs, MMP-11,12, 3, 8. Is there any changes in TGF-beta?

·        The authors have used only one commercial cell line. Please show the readouts in additional cells lines, preferably primary human epithelial cells.

·        Since there is a change in MUC5AC, please measure IL-13 since it is the major inducer of MUC5AC.

·        What about IL-1beta changes as it is a major marker of inflammation? Please measure it.

Comments on the Quality of English Language

Please change the typographical errors and increase/improve axes sized and readability. 

Author Response

Reviewer 2

The authors study the role of different flowering stages of Perilla frutescens in inhibition of PM2.5 induced oxidative stress, inflammation, mucus production and fibrosis. Firstly, the authors need to pay extensive attention to the typing errors and components of the manuscripts. For example, legends are not matching with Figures. This is extremely confusing as readers. Please screen the manuscript for grammatical errors and typographical errors. The authors have performed extensive experiments, but the poor visibility of the figures, and usage of one commercial cell line to measure the limited readouts (ROS, NO, MDA, TNF-a, IL-6, MUC5AC and MMP-9) reduces the enthusiasm of the study.

  1. Please increase the size and readability of the axes in the Figures 2,3 and 4, 7, 8. It is very hard to read the figures.

→ Thank you for your advice. I have revised the figures by increasing the size of the axis labels to improve overall readability.

  1.  Please remove one of the two Figures 2 and 3 since they convey the same information.

→ Thank you for your comments. Figure 3 presents essential results as it allows for the comparison of differences between cultivars, unlike Figure 2. However, since the reviewer seems concerned about redundancy with Figure 2, I will move Figure 3 to the Supplementary Data section.

  1.  Please show the results from the screening that was performed to determine the dosage of PM2.5, at least in supplemental.

→ Thank you for your advice. I have added the results on cell viability based on PM2.5 concentration as supplementary data.

  1. Please correct/update Figure 5C. The first bar shows sample (-) and PM2.5(+) with almost 100% viability. But second bar also shows sample (-) and PM2.5(+) with almost 50% viability. How is that possible?

→ Thank you for your advice. The first bar represents sample (-) and PM2.5(-), while the second bar represents sample (-) and PM2.5(+). There was an error in the labeling of the figure, and I have corrected it.

  1. Please re write the legends properly. For example, the authors indicate write that Figure 5 B and C demonstrates changes in NO and MUC5AC, but Fig 5 only shows cell viability data. Similarly, the legends for Fig 5 and 6 are exactly same, but Fig 6 is NO and MUC5AC. As authors of scientific manuscripts, it is important to pay attention to what is written in the legends and if that corresponds to the Figures.

→ Thank you for your advice. I have revised the legends for Figures 4 and 5 accordingly

  1.  Please improve the font size and readability of Fig 7. I was not able to read it at all.  There is no point is putting figures if no one can read it.

→ Thank you for your advice. I have made efforts to improve the readability of Figure 6 by adjusting the resolution and the font size of the subcaptions in the correlation analysis

  1. In Fig 8, the authors state “mechanism” in line # 548 and 549. However, this is not a mechanism, it is pharmacological readouts. Please change the word “mechanism”.

→ I agree that it is difficult to fully explain the mechanism with the current data. Therefore, I have revised the description of the mechanism. In future studies, I plan to conduct additional research using Western blot and RT-PCR with the optimal extracts to elucidate the mechanism of action. This study is significant in determining the optimal conditions for secondary metabolite production and bioactivity

  1. Elaborate the reason for measuring NO and why is this data relevant to the study?

→ PM2.5, which was used as a stimulant in this study, can induce inflammation in human airways and lung cells. Therefore, measuring the secretion of nitric oxide (NO), a key marker of inflammation, and assessing the inhibition rate of NO is crucial

  1. The authors state this flower can protect against fibrosis, and just indicate the levels of MMP-
  2. Firstly, the role of MMP-9 is unclear, as in whether it is pro or anti-fibrotic, making it a weak candidate for evaluation of fibrosis. I think the authors need to demonstrate additional fibrosis marker changes, like TIMPs, MMP-11,12, 3, 8. Is there any changes in TGF-beta?
  3. Since there is a change in MUC5AC, please measure IL-13 since it is the major inducer of MUC5AC.

→ (Answer to 9, 10, 12) This study, as previously explained, focuses on extensively investigating the changes in secondary metabolites according to various growth stages of perilla flower clusters and how these changes affect bioactivity before conducting mechanistic studies on their respiratory health benefits. Therefore, we first analyzed the key markers related to inflammation, mucus hypersecretion, and fibrosis across various samples. Among the fibrosis markers, we prioritized the analysis of MMP-9, which is one of the most representative indicators of fibrosis. In future studies, to elucidate the mechanism, we plan to conduct expression analyses of various MMPs, TGF-beta, and collagen using Western blot and RT-PCR techniques.

  1. The authors have used only one commercial cell line. Please show the readouts in additional cells lines, preferably primary human epithelial cells.

→ Before conducting this experiment, we performed a preliminary study by treating fine dust and perilla extract on upper respiratory tract cells, lower respiratory tract cells, and lung cells. Among these, the upper respiratory tract cells were found to be the most suitable for the experiment, and thus, we selected them for the main study. However, to investigate the effects of perilla extract on the entire respiratory system, we are currently conducting in vivo experiments using a mouse model stimulated with fine dust. We plan to report the overall effects on the respiratory system in future studies.

  1. What about IL-1beta changes as it is a major marker of inflammation? Please measure it.

→ I agree that analyzing IL-1beta along with TNF-alpha and IL-6 as key markers of inflammation is important. Therefore, we conducted an additional analysis using an ELISA kit and included the results in Figure 7 (I)

Round 2

Reviewer 1 Report

Comments and Suggestions for Authors

The authors have addressed almost all my requests and improved the manuscript's quality considerably. However, some minor issues need to be fixed before the Editor's final decision.

1. Double-check the Figure numbering after the previous Figure 3 is moved to supplementary files.

   2. The conclusion still needs to be revised. A conclusion should report the study's major findings and cannot be written as an introduction. Delete the starting sentences and begin the sentence in Line 586.

“This study revealed the variation in phenolic acids, flavonoids, and polycosanols in perilla flowers and seeds at different post-flowering developmental stages, and assessed …”

Line 588. You omitted “40 days’

Line 590. Delete “overall”

Line 592. Delete the sentence “Identifying …… periods”

3. There are different types of perilla, depending on leaf and seed colors. Please provide more information about the phenotypic characteristics of the five varieties in materials and methods.

Comments on the Quality of English Language

Minor editing required.

Author Response

Point-by-point responses to the Editor’s and Reviewers’ comments

Reviewer Comments:

Reviewer 1

The authors have addressed almost all my requests and improved the manuscript's quality considerably. However, some minor issues need to be fixed before the Editor's final decision.

  1. Double-check the Figure numbering after the previous Figure 3 is moved to supplementary files.

Thank you for your advice. We have checked and revised the Figure numbering throughout the manuscript.

  1. The conclusion still needs to be revised. A conclusion should report the study's major findings and cannot be written as an introduction.

Delete the starting sentences and begin the sentence in Line 586.

“This study revealed the variation in phenolic acids, flavonoids, and polycosanols in perilla flowers and seeds at different post-flowering developmental stages, and assessed …”

Line 588. You omitted “40 days’

Line 590. Delete “overall”

Line 592. Delete the sentence “Identifying …… periods”

Thank you for your advice. I have revised conclusion according to your advice

  1. There are different types of perilla, depending on leaf and seed colors. Please provide more information about the phenotypic characteristics of the five varieties in materials and methods.

Thank you for your advice. I have added cultivar information for the five materials.

Reviewer 2 Report

Comments and Suggestions for Authors

The authors have provided satisfactory revision. 

Author Response

Dear Reviewer,

Thank you very much for your valuable feedback and thoughtful suggestions on our manuscript. Your insights have been instrumental in improving the quality of our work. We have carefully addressed each of your comments and believe they have greatly enhanced the clarity and depth of our research.

We appreciate the time and expertise you have contributed to reviewing our paper.